# Osteoderm Development during the Regeneration Process in *Eurylepis taeniolata* Blyth, 1854 (Scincidae, Sauria, Squamata)

**DOI:** 10.3390/jdb11020022

**Published:** 2023-05-24

**Authors:** Gennady O. Cherepanov, Dmitry A. Gordeev, Daniel A. Melnikov, Natalia B. Ananjeva

**Affiliations:** 1Department of Vertebrate Zoology, Faculty of Biology, Saint Petersburg State University, 199034 St. Petersburg, Russia; cherepanov-go@mail.ru; 2Department of Biology and Bioengineering, Institute of Natural Sciences, Volgograd State University, 400062 Volgograd, Russia; dmitriy8484@bk.ru; 3Zoological Institute of RAS, 199034 St. Petersburg, Russia; persicus@zin.ru

**Keywords:** *Eurylepis taeniolata*, osteoderms, development, anatomy, histology, regeneration

## Abstract

Osteoderms are bony structures that develop within the dermal layer of the skin in vertebrates and are very often found in different lizard families. Lizard osteoderms are diverse in topography, morphology, and microstructure. Of particular interest are the compound osteoderms of skinks, which are a complex of several bone elements known as osteodermites. We present new data on the development and regeneration of compound osteoderms based on the results of a histological and Computed Microtomography (micro-CT) study of a scincid lizard: *Eurylepis taeniolata*. The specimens studied are stored in the herpetological collections of the Saint-Petersburg State University and Zoological Institute of the Russian Academy of Sciences located in St. Petersburg, Russia. The topography of osteoderms in the integuments of the original tail area and its regenerated part was studied. A comparative histological description of the original and regenerated osteoderms of *Eurylepis taeniolata* is presented for the first time. The first description of the development of compound osteoderm microstructure in the process of caudal regeneration is also presented.

## 1. Introduction

Osteoderms (ODs) are bony structures that develop within the dermal layer of skin in vertebrates. They are present in most major extant tetrapod lineages, such as amphibians (anuran), mammals (armadillo), and reptiles (turtles, crocodilians, and squamates). They are also found in many extinct lineages, such as early tetrapods and dinosaurs. Within squamates, they are absent in amphisbaenians; however, they are well developed in 12 lizard families: Scincidae, Cordylidae, Gerrhosauridae, Anguidae, Lacertidae, Helodermatidae, Xenosauridae, Shinisauridae, Lanthanotidae, Anniellidae, some Varanidae and Gekkota [1,2,3,4,5,6,7,8,9]. Within geckos (Gekkota), ODs are uncommon and were previously only described in the genera *Gekko* (*G. gecko*, *G. reevesii*), *Tarentola* and *Geckolepis* [1,10]. This year, the first description of dermal armour in snakes in the Erycinae subfamily (*Eryx conicus* and *E. colubrinus*) was published [11].

ODs are highly polymorphic, having a vermiform shape in varanids; a flat, imbricating, squamous shape in anguids, scincids and cordylids; and a non-imbricating granule, plate, or bead-like form in xenosaurids, lanthanotids, shinisaurids and helodermatids. Such variability is observed even within a single species [1,2,12,13]. The ODs are separated from each other via connective tissue hinges, which are formed of dense oriented collagen fibers, allowing relative mobility [14].

In some species, the outer surface (closest to the epidermis) of the ODs is covered by an enigmatic capping tissue [4,6,15,16,17]. This tissue, which is termed osteodermine [15], is a dense, avascular, cell-poor and hypermineralized enamel-like layer that lacks intrinsic collagen [15,16,18]. The so-called osteodermine tissue is described in particular in helodermatids, the geckos *Tarentola annularis* and *T. mauritanica*, and at least one fossil glyptosaurine anguid [6,8].

The structures observed in *Scincus scincus* and *Eumeces schneideri* correspond to this description, as they seem acellular and have overlapping collagen dense bone tissue [14]. The degree of osteodermine is different even among closely related taxa: the histological sections of ODs in scincid *Corucia zebrata* show only secondary osteons, but no osteodermine [8].

ODs may be completely absent; alternatively, they can be developed only on the head or dorsum, or all over the lizard’s body. Modern reviews [5,8] that summarized the make-up of these structures, their evolutionary transformations and prospects for further research noted that ODs are recorded in many taxa of lizards (up to 25% of species); however, their development is not necessarily registered even within the members of the same genus (for example, *Varanus*). The study of the form and function of these dermal ossifications is extensive, beginning with the classical morphological literature [9,12,19,20,21,22], but certainly does not cover the entire diversity of extant lizards, including the vast family of skinks, which contains about 1750 species. Camp [12] indicates the presence of ODs in lizards of the genera *Scincus*, *Gongylus*, *Seps*, *Lygosoma*, *Mabuya* and *Acontias*. Morphological characterization and biomimetic design of ODs were described in *Egernia* [13] and *Corucia zebrata* [7]. More attention was paid to the study of head ODs, often in the framework of modern studies of cranial morphology using micro-CT [2,8]. At the same time, developmental sequence of ODs and their formation at various stages of ontogeny and, in particular, the morphological and histological characteristics and development of caudal ODs remain poorly studied [5].

Of particular interest are complex ODs within the ecologically diverse members of the skink family. The main goal of the present study is the development of ODs during tail autotomy and regeneration, which is typical for species of the scinidae. The data obtained and the analysis herein will make it possible to supplement the existing hypotheses regarding both the caudal regeneration process itself and the ODs’ involvement in this process.

In this study, the previously unexplored genus and species Ribbon-sided Skink, *Eurylepis taeniolata* Blyth, 1854, was used. We provide the first histological description of regenerated and non-regenerated compound ODs, as well as new information about the growth dynamics of regenerated ODs and the development of the osteodermal microstructure.

## 2. Materials and Methods

The study was carried out based on collections of materials stored at the St. Petersburg State University (SPbU) and the Zoological Institute, Russian Academy of Sciences (ZISP). The St. Petersburg State University material was fixed in 4% formalin, while the ZISP specimens were preserved in ethanol. Histological and morphological examination was carried out with specimens owned by the St. Petersburg State University (SPbU number 1773-1777, 1779-1782, R-125, R-150, R-154-158), and the tomographic study used specimens of the Zoological Institute (ZISP numbers 18967: 5334, 5327, 5328. A total of 16 SPbU specimens of tail regenerates from 3 to 62 mm in length were studied. Nine samples of regenerating tails were selected for histological sectioning. This material was washed free of formaldehyde in running water, decalcified in 5% nitric acid for 30 days, passed through alcohols and chloroform and embedded in paraffin blocks. The blocks with embedded tissue were cut in the transverse and longitudinal planes into 10 µm-thick sections on a microtome. Histological sections were stained according to the standard method with Delafield’s hematoxylin/eosin or azan/azacarmine, according to Heidenhain. This staining method was used to visualize the structure of the collagen matrix. The total amount of histological material includes 64 preparations (microscope slides) with 6 to 20 sections on a slide (Table 1).

Several samples (numbers R-125, R-150, R-154-158) were stained with alizarin red and cleared in glycerin following the method of Hardaway and Williams [23]. All of the above materials are stored at the St. Petersburg State University (St. Petersburg, Russia). Microscopic examination was carried out using a Leica DM 2500 microscope. Microphotographs of morphological sections were taken using a Leica DFC 450 C camera (user interface was The Leica Application Suite). Samples stained with alizarin were photographed with a Canon Power Shot SX130IS camera. The drawings were made using Adobe Illustrator CS5.1.

The anatomical structure of the tails was studied via the micro-CT method (Center of Collective Use, Zoological Institute Russian Academy of Sciences, Saint Petersburg (https://www.ckp-rf.ru/ckp/3038/, accessed on 5 March 2023) using a Neoscan N80 desktop scanner (Neoscan BVBA, Mechelen, Belgium, S/N = 80B2106, Software Version = 2.2.5) equipped with a 7 Mp Flat-panel camera. The tails of numbers ZISP 18967: 5328, 5327, 5334 were scanned (see Appendix A). The specimens were placed in a plastic vessel and scanned sequentially. Scanning was performed with rotation step 0.200° and with no averaging and no 360° rotation; other parameters are specified in Table 2.

The study of ZISP specimens showed that autotomy and regeneration occur in both males and females of *E. taeniolata*, and in individuals of various sizes (body length): adult (122.8 mm), subadult (81.7) and juvenile (55.9). Tail shedding occurs both in the proximal (ZISP 5328: Lcd 81.7: 31.5 + 50.2) and distal (ZISP 5327: Lcd 122.8: 104.0 + 18.8, 5334: Lcd 55.9 49.8 + 6.1) half of the tail (Lcd means the tail length consisting of two parts: original and regenerated).

The resulting subscan data arrays were connected vertically to obtain the general tomogram. Data were processed using the original Neoscan reconstruction software and the NRecon, Bruker (Program Version 1.7.4.6), as well as for visualization of 3D models CTVox (Bruker, Billerica, MA, USA).

## 3. Results

The field observations compiled by the first author made it possible to conduct observations on tail regeneration, for which 11 large specimens were selected and numbered (Figure 1). The selected individuals were housed in a terrarium under standard conditions; regeneration was observed for one year. We showed that the rate of regeneration and size characteristics of tail regenerates depend on the individual characteristics of lizards. Thus, the length of the tail regenerates in individuals 1, 4, 5 and 8, which develop over almost the same period of time (9.5–10 months), ranges from 12.5 to 49 mm. On the other hand, tail regenerates of equal size in individuals 10 and 11 (5.5 mm) differ fourfold in terms of development time, being 2 and 8 months, respectively. There is a tendency toward a decrease in regenerative potency during the secondary autotomy and the formation of a secondary regenerate (second generation). This trend is clearly demonstrated in specimen number 2 with coeval 6-month-old first and second regenerates, which were 28 and 5 mm long, respectively. In specimen 10, a 7.5-month-old regenerate of the second order was shorter than a two-month-old regenerate of the first order.

### 3.1. The Structure of the Original Osteoderms

The ODs of *E. taeniolata*, which are located in the dermis under the epidermal scales, are well developed and cover almost the entire body of the skink with a continuous “armor”; however, the metacarpuses and metatarsus of the extremities remain partially free, and the phalanges of the fingers are completely devoid of them (Figure 2). ODs of *E. taeniolata* are located in the papillae of horny scales, which overlap each other akin to tiles. Their number is always the same as the number of scales. ODs also cover the regenerated tail; its calcified rod is not visible on tomography, though mineralization is evident on one the alizarin preparations.

Seen from above, the ODs are oval in shape and transversely expanded (Figure 3). They are located in the skin with an overlap: the anterior ODs overlap the adjacent posterior and posterolateral ODs. The overlap zone is approximately one-third of the area of the OD. ODs have a complex structure. Each OD consists of several small bony plates—osteodermites—arranged in two rows: anterior and posterior. Osteodermites, when viewed from above, are roughly rectangular in shape, and are elongated along the anteroposterior axis. They are separated via narrow zones of non-mineralized tissue, i.e., the interosteodermitic hinges. On the dorsal side of the tail, the number of osteodermites of each OD usually varies from 10 to 12, while on the ventral side, they can number up to 20 or 21.

As a rule, there are 1–2 fewer anterior osteodermites than posterior osteodermites. In some cases, in the central part of the OD, small intercalated ossifications (intermediate osteodermites) may be present between the anterior and posterior rows of osteodermites. The outer surface of an osteodermite is perforated by between 2–3 and 10 neurovascular openings. Most often, they are located in the middle of the osteodermite as a longitudinal row of openings.

ODs lie in the skin at the boundary between the compact and superficial layers of the dermis and are located a small distance from the epidermis; they underlie almost the entire area of the outer surface of horny scales (Figure 4a). The ODs are thin: the maximum thickness does not exceed 150 μm, while the ratio of thickness-to-length of the OD is 1/30. In the OD, only two layers are clearly distinguishable, which are represented by the basal (basal lamina) and superficial cortex (Figure 4b). The basal cortex is up to two-thirds the thickness of the OD. It is composed of compact bony tissue. The fibrous matrix of the basal cortex is not homogeneous within one OD (and one osteodermite) in its different parts (Figure 4d). In the thick central part of the OD, the matrix has a woven bony structure. At the edges, the OD is relatively thin and is composed of layered mineralized fibers—fibrolamellar bone. The basal cortex is connected to the underlying skin via bundles of collagen fibers (Sharpey’s fibers), which are deeply rooted in the bone and extend into the compact layer of the dermis. The superficial cortex does not have thick bundles of fibers, instead including thin fibrils that emerge perpendicularly on the surface of the OD.

The superficial cortex is characterized by the presence of osteodermin, which is the hypermineralized tissue (Figure 4b,c). Osteodermine is characterized by the absence of internal collagen and osteocyte cavities, with noticeable concentric growth lines. Osteodermine is localized largely in the region of the hinges between osteodermites. ODs are perforated with vertical canals for blood vessels and nerves that exit to the outer surface of the bone. Sometimes, at the border between the basal and superficial cortex, the internal vascular system is visible in the form of horizontal canals (Figure 4c). Accumulations of osteocytes are also observed here. In the area of contact between the osteodermites that form the OD, there are large and clearly organized bundles of Sharpey’s fibers in the dermal compact layer. In areas that extend inside the OD, Sharpey’s fibers are mineralized (Sharpey-fiber bone).

### 3.2. Development of the Integument during Regeneration and Zoning of the Tail Regenerate

Three zones can be distinguished in the developing tail regenerate, which reflect successive stages of integument formation (Figure 5a). These zones are the zone of undifferentiated integument, the zone of scale regeneration and the zone of OD regeneration.

Zone of undifferentiated integument: This narrow zone covers the blastema region and a small space near it (Figure 6a). The epidermis is slightly keratinized. In the area of the blastema, it forms up to 10 layers of differentiating cells and includes several layers of keratinized cells that are shed (Figure 6b). In the proximal part of this zone, the epidermis is noticeably thickened; its germinative stratum consists of 15 to 20 cell layers. The cells of the basal layer of the epidermis are irregularly shaped.

The tissue located under the epidermis is characterized by a dense accumulation of mesenchymal cells and a high number of blood vessels and cavities filled with blood cells. A significant concentration of melanocytes is observed in the apical zone of the blastema under the epidermis, which is opposite the emerging caudal end of the cartilaginous tube. The distal tip of the tail is characterized by the early differentiation of internal structures: ependymal tube, cartilaginous tube, surrounding adipose tissue and muscles.

Zone of scale regeneration: Depending on the length of the tail regenerate, this zone can cover between 5 and 10 segments (forming rows of scales) (Figure 6c). The germinative layer of the epidermis is thinner than the epidermis of the blastema region. The horny layers, in contrast, are thickened and form a keratinized plate on the surface of the regenerate. The cells of the basal layer of the epidermis are cuboidal. The basement membrane is well defined. The dermis is stratified into a deep compact layer and a superficial layer (stratum laxum). The compact layer is characterized by the presence of a dense collagen matrix. The superficial layer bears relatively thin and loosely arranged collagen fibers; numerous blood vessels are concentrated here.

The formation of scales begins as a local thickening of the epidermis and the immersion of this area deeper into of the dermal layer of the skin in the form of a peg-shaped structure (Figure 6d). The area of epidermal invaginations is characterized by the presence of dermo-epidermal anchor filaments closely associated with the basement membrane. Eventually, invaginations of the epidermis take the form of asymmetric pockets oriented in the dermis in the anterio-proximal direction. In the interspaces between invaginations, dermal papillae of future scales and a system of their blood vessels are formed. The epidermal pockets submerge to the level of the compact dermis, where a hinge region is formed between the scales. Gradual stratification of the horny layers of the epidermis in the area of epidermal pockets leads to the formation of the outer and inner surfaces of the scales.

Zone of OD regeneration: ODs cover the regenerated tail, in the structure of which the calcified rod is not visible on tomography (Figure 5f); however, mineralization is recorded on the sections of the alizarin and histological preparations (Figure 6g).

The zone of OD regeneration varies greatly in length depending on the length of the tail regenerate and the level of its differentiation. In the early stages of its formation, the tail regenerate can be composed of only one or two rows of regenerating ODs; in the final stages, it is a continuous osteodermal cover of the regenerated tail (Figure 2 and Figure 5b–e). The epidermis is scaly, keratinized and exhibits a thick beta-keratin layer. The dermis is three-layered, with a well-defined superficial (papillary) layer, a compact layer and a hypodermis (Figure 6f). The superficial layer is characterized by a loose structure and the absence of large bundles of collagen, which are characteristic of the compact layer. In the compact layer, collagen bundles are oriented predominantly along the longitudinal axis. The hypodermis is distinguished through the transverse orientation of the fibers. ODs are built on the border of the compact and superficial layers of the dermis. Each OD develops from several distinct centers of ossification—the osteodermite anlagens—located along the periphery of the dermal papilla of the horny scales (the regeneration of ODs is described in detail below).

### 3.3. Regeneration of Osteoderms

Stage I: At the earliest stage of development, the anlagens of ODs are small local aggregations of collagen fibers surrounded by fibroblasts (Figure 7a,b). In the dermis, these anlagen are located along the upper boundary of the compact layer; they are separated from the epidermis via a relatively thin superficial layer of the dermis. The superficial layer is distinguished from the compact layer via a looser structure and the absence of large bundles of collagen fibers, which are characteristic of the compact layer (Figure 7b). The fibrous anlagens of osteodermites of the anterior row of the composite ODs appear first. They lie near the bottom of the epidermal pockets, i.e., in the hinge area between regenerating scales. The anlagens of osteodermites of the posterior row appear somewhat later with a lag of two or three rows (Figure 5a). These elements also lie on the boundary of the compact and superficial layers of the dermis in the middle part of the dermal papilla of regenerating scales.

Stage II: Ossification of osteodermites begins with the mineralization of the collagen fibers in their anlagen. As a result, an osseous lamina (osteoid) is formed, surrounded by osteoblasts and containing a small number of internal osteocytes (Figure 7c). The sequence of ossification of osteodermites corresponds to the order in which they appear during regeneration: the anterior row of osteodermites is first ossified, followed by the posterior osteodermites (Figure 7d). At this stage, all osteodermites of the regenerating ODs are separated by wide spaces of non-mineralized dermis. When viewed from above, they have a rounded or oval shape. Short trabeculae, which can be interpreted as evidence of peripheral growth, form along the edges of osteodermites. The growth of osteodermites reduces the spaces between them. At this stage, the OD has the appearance of an oval ring of ossified elements located along the periphery of the scale (Figure 5a). The number of osteodermites within one OD varies from 5 to 20 and correlates with the size of the scale in which it was formed.

Stage III: At this stage of regeneration, small tubercles are formed on the outer superficial layer of the primary lamina, growing in the superficial layer of the dermis (Figure 8a). Thus, the osseous anlagen of ODs acquire a two-layered structure, which includes the basal cortex and the superficial tubercular layer. The basal cortex is organized based on the mineralization of the collagen matrix of the compact layer of the dermis. It maintains a rigid connection with the underlying dermis anchored by Sharpey’s fibers. The superficial layer is distinguished through the presence of a large number of osteocytes enclosed in bony tissue. This layer also contains thin collagen fibers of the superficial dermis. The latter are loosely arranged and oriented along the vertical axis stretching from the outer surface of the ODs to the epidermis. Osteoblasts surround the bony elements of regenerating ODs. Osteodermites show considerable growth along the periphery, are almost in contact with each other (Figure 5a) and are nearly quadrangular in shape. The osteodermites of the anterior row are noticeably larger than are those of the posterior row.

Stage IV: The regeneration process ends once the osteodermal cover is fully formed. However, the regenerated ODs do not reach the full size of the original ones (Figure 5b). The maximum thickness of the OD varies within 100 μm; along the edges, the OD becomes thinner. In the central part, the ODs have a three-layered structure (Figure 8b); on the periphery, the structure has only two layers. The basal cortex is composed of a fibrous-lamellar or woven bony tissue; osteocyte lacunae are very rare. This layer may account for more than half of the ODs’ overall thickness. In the basal cortex, two zones are clearly distinguished: the lower, younger and less mineralized zone, and the upper, more mature zone. Thick Sharpey’s fibers inextricably link the basal cortex to the collagen matrix of the underlying dermis (Figure 8c). The superficial cortex has a homogeneous bony structure, probably containing only very thin fibers extending from the superficial layer of the dermis. In the thicker areas of the ODs, an intermediate bony layer is distinguished between the basal and superficial cortex. It is characterized by the presence of a large number of osteocytes and a system of thin intertwined canals. In some areas, the ODs are permeated with vertical neurovascular canals that open onto their surface. There is a concentration of fibroblasts and Sharpey’s fibers, which form hinges in the areas of contact between osteodermites. On the surface of the ODs, the periosteum is clearly expressed and contains fibrous structures and groups of osteoblasts. The formation of osteodermine is found in some areas of the superficial cortex (Figure 8d).

The trend of the placement of ODs in the zone of their regeneration differs in individuals of different age groups. In an adult specimen (ZISP 5328) with fully formed ODs on the original part of the tail, a continuous osteodermal cover developed (Figure 5b,c). In subadults (ZISP 5327), on the regenerate between adjacent ODs, areas without osteodermites are clearly visible (Figure 5d,e). A juvenile specimen (ZISP 5334) exhibits an interesting case of the rate of formation and regeneration of ODs. On the intact part of the tail, osteodermites have not yet completed the formation of a continuous cover (areas without osteodermites are clearly visible, and osteodermites are not combined into ODs), while the size of these ODs is smaller than regenerating ones (Figure 5f). This fact may indicate that, in this species, the regeneration of ODs proceeds at a faster rate than their formation on the intact region of the tail.

## 4. Discussion

Scincoidea, scincid lizards and, in particular, *Scincus scincus* and *Eumeces schneideri*, have a specific pattern of ODs referred to as “compound ODs” [14]. In lizards of other families, ODs are singular elements [1,3,8,11,16]; scincid lizards have ODs composed of several smaller elements, known as osteodermites, linked together via fibrous joints [2].

In contrast to the species mentioned above, in the specimens of *E. taeniolata*, osteodermine-like tissue is found on the surface of the original (Figure 4); in all probability, this tissue is also present on the regenerating ODs (Figure 8d). We noted that in *E. taeniolata*, ODs are compound elements consisting of multiple joined osteodermites, which is typical for the scincid lizards previously studied. The number of osteodermites in the original caudal ODs ranges from 10 to 18 (Figure 3). Regenerating ODs are more variable in their composition. The number of osteodermites varies from 5 to 20, depending on the size of the horny scale, in the area in which the OD is formed (Figure 5). Small intercalated osteodermites, either round or oval in shape, are often present between the rows.

The regenerative development of *E. taeniolata* ODs is characterized by the formation of several separate centers of ossification, i.e., the anlagens of osteodermites (Figure 5a). Firstly, ossifications form on the anterior row of the future complex OD, followed by the posterior row. The anterior elements are laid down in the hinge area between adjacent epidermal scales; the posterior elements develop closer to the apex of the dermal papilla of the scale (Figure 7d). In the early stages of development, the osteodermite anlagens are separated by a wide expanse of non-mineralized dermis. The gradual growth of osteodermites leads to their convergence and the emergence of contact between them. Thus, interosteodermitic hinges are formed between the adjacent elements of the OD. In the regions of these hinges, the osteodermites are connected through bundles of unmineralized Sharpey’s fibers anchored in their bony tissue (Figure 8b).

Based on the study of tail regeneration in *E. taeniolata*, we succeeded in tracing the complete development cycle of its osteodermal cover. This method allowed us to identify four successive stages of dermal ossification and describe the formation of the microstructure of regenerating ODs.

Stage I: Appearance of dermal anlagen of osteodermites as the aggregations of collagen fibers at the boundary between the compact and superficial layers of the dermis (Figure 7a,b).

Stage II: Formation of centers of ossification within the dermal anlagens of osteodermites resulting from the mineralization of the collagen matrix (Figure 7c,d).

Stage III: Formation of a two-layer structure of bony plates: basal cortex and upper tubercular layer (Figure 8a). The basal cortex grows centripetally with the inclusion of fiber bundles of the compact layer of the dermis; the tubercular layer grows centrifugally in the superficial dermal layer with the inclusion of numerous osteocytes in the bone structure.

Stage IV: Formation of a three-layer bone structure: basal cortex an intermediate layer (transformed tubercular) and the superficial cortex (Figure 8b,d). The superficial cortex has a homogeneous structure of bone tissue and includes “islands” of osteodermine.

In the specimens of *E. taeniolata* we studied, as in most other skinks, complex ODs are large and overlap each other. They do not form rigid contacts with each other as the Sharpey-fibered articulation. However, their constituent osteodermites have such Sharpey’s fibrous joints (interosteodermitic hinges). In their morphological and histological properties, osteodermites resemble small individual ODs, as shown in the geckos *Tarentola annularis* [8], *T. annularis* and *T. mauritanica* [16]. Thus, there is reason to believe that the evolutionary origin of complex skink ODs is related to the association of several small ODs into a single complex. Perhaps the explanation for this association lies in the formation of large overlapping epidermal scales in the ancestors of skink lizards, i.e., wide morphogenetic zones, allowing them to combine several centers of ossification.

A further study of the osteodermal cover, as well as its ontogenetic development and regeneration, will reveal differences in lizards within the diverse and numerous (about 1750 species) scincid family and its seven subfamilies (Acontinae, Egerniinae, Eugongylinae, Lygosominae, Mabuyinae, Sphenomorphinae, Scincinae), reflecting the phylogenetic diversity and distribution of evolutionary lineages.

## 5. Conclusions

Skinks have unique compound ODs, in which each OD is a complex of several bone elements, known as osteodermites, which are interconnected through narrow non-mineralized zones of Sharpey’s fibers. A comparative histological and micro-CT study of the original and regenerated ODs of *E. taeniolata* showed the fundamental similarity of their morphological and microstructural organization. It was shown that ODs have a complex microstructure, comprising various types of bony tissue, such as lamellar bone, parallel-fibered bone, woven bone and Sharpey-fiber bone. The presence of osteodermine in the superficial cortex of ODs was revealed. The process of development of complex ODs during the regeneration of the tail of *E. taeniolata* after autotomy was described. As a result, four stages of regenerative osteogenesis were identified, ranging from the appearance of a fibrous anlage to the formation of a three-layered bony structure. The similarity of osteodermites of skinks and small ODs of geckos [8,16] made it possible to hypothesize that skink compound ODs arose as a result of the association of several small ODs into a single complex. The lack of data on the ontogenetic development of the integument in skink lizards did not allow for a comparative analysis of the ontogenetic and regenerative development of complex ODs. However, such an analysis would be extremely interesting, especially if one bears in mind that the embryogenesis and regeneration of most of the anatomical organs of lizards (in particular, the axial skeleton, muscles and scales) are fundamentally different. These are aspects we aim to address in future research.

## Figures and Tables

**Figure 1 jdb-11-00022-f001:**
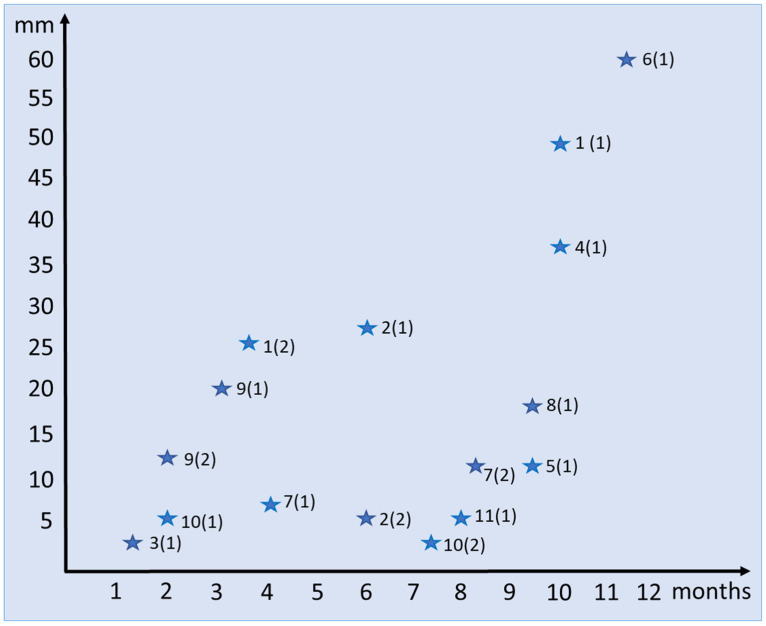
Diagram of relationship between size (mm) and age (month) of regenerating tails in *E. taeniolata*. First number indicates number of individuals, in parentheses—number of generation (regenerate of first or second order).

**Figure 2 jdb-11-00022-f002:**
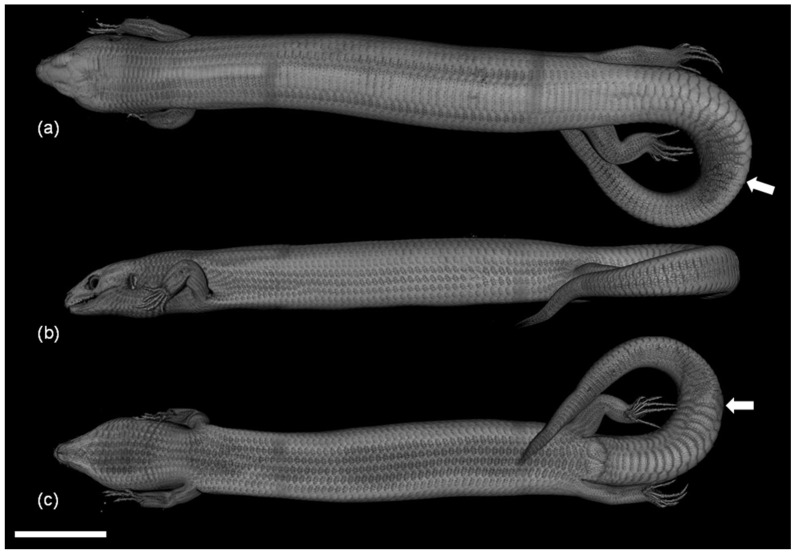
Spatial organization of osteodermal cover of *E. taeniolatus* (R-5328-2) according to results of three-dimensional micro-CT reconstruction: (**a**) dorsal view; (**b**) lateral view; (**c**) ventral view. White arrows show area of autotomy. Scale bar: 20 mm.

**Figure 3 jdb-11-00022-f003:**
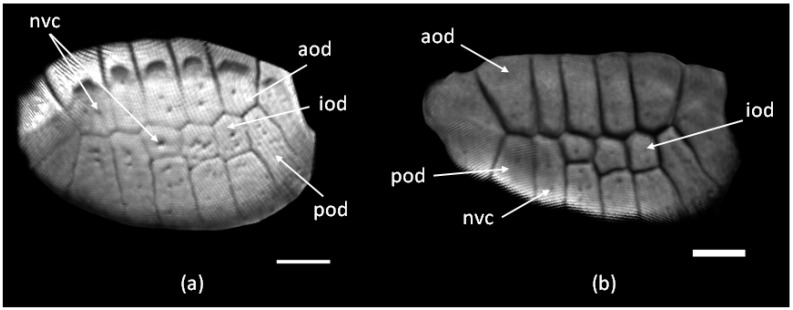
Original tail osteoderms of *E. taeniolata* (ZISP # 5328) based on 3D micro-CT reconstruction: (**a**) outer and (**b**) inner surfaces. Abbreviations: aod, anterior osteodermites; iod, intermediate osteodermites; nvc, neurovascular canals; pod, posterior osteodermites. Scale bars: 2.5 mm.

**Figure 4 jdb-11-00022-f004:**
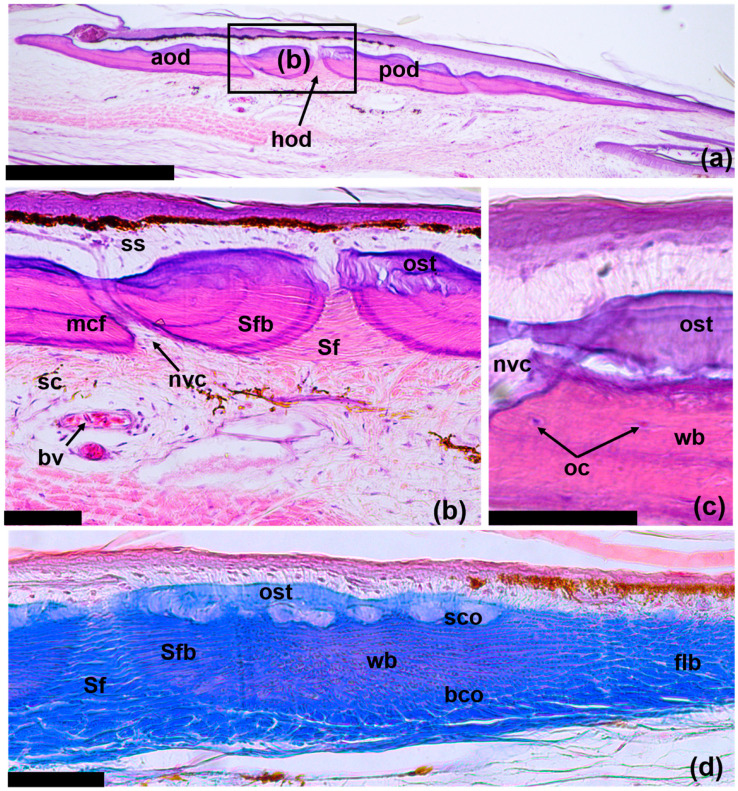
Longitudinal sections of original tail osteoderms of *E. taeniolata*: histological preparations (**a**,**b**) SPbU # 1773; (**c**) SPbU # 1774; (**d**) SPbU # 1782. Abbreviations: aod, anterior osteodermite; bco, basal cortex; bv, blood vessel; flb, fibrolamellar (parallel-fibered) bone; hod, hinge region between osteodermites; nvc, neurovascular canal; oc, osteocytes; ost, osteodermine; pod, posterior osteodermite; sc, stratum compactum; sco, superficial cortex; Sf, Sharpey’s fibers; Sfb, Sharpey-fiber bone; ss, stratum superficiale; wb, woven bone. Sections are stained with Delafield’s hematoxylin, eosine (**a**–**c**) and with Azan, azo-carmine B (**d**). Scale bars: (**a**) = 500 µm, (**b**–**d**) = 50 µm.

**Figure 5 jdb-11-00022-f005:**
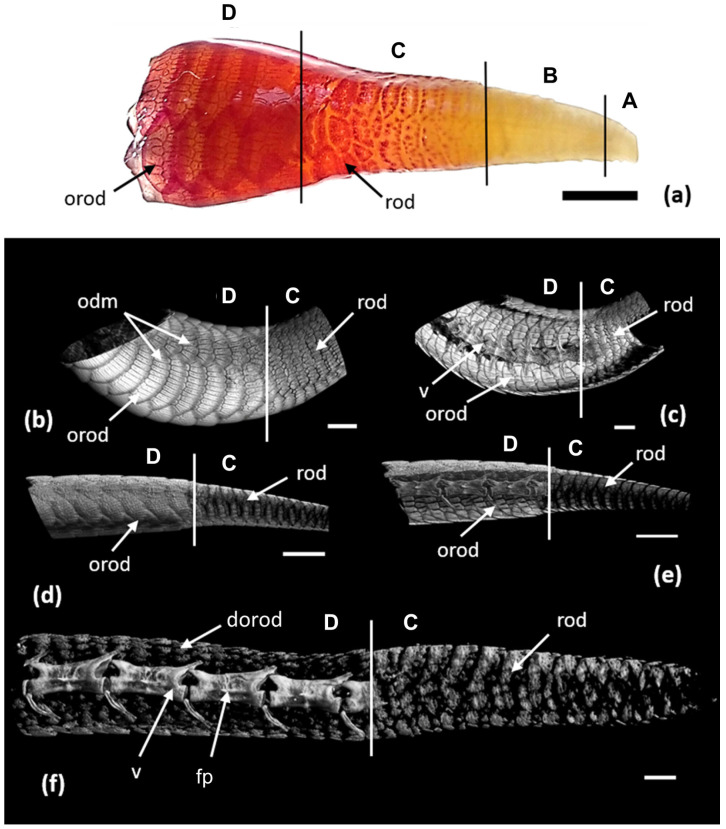
Spatial organization of osteodermal integument of tail of *E. taeniolata*: (**a**) alizarin preparation (SPbU # R-125) with a regenerate 17.5 mm long (9.5-month-old); (**b**–**f**) 3D micro-CT reconstruction of tail with regenerate: (**b**) ventral side and section in frontal plane; (**c**) tail of an adult specimen (ZISP # 5328); (**d**) left side and section in sagittal plane (**e**) of the tail of a subadult specimen (ZISP# 5327); (**f**) Left side of tail of a juvenile specimen (ZISP#5334). Abbreviations: dorod, developing original osteoderm; fp, fracture plane; odm, osteoderms; orod, original osteodermites; rod, regenerating osteodermites; v, vertebra. Vertical lines show key areas: A, zone of undifferentiated integument; B, zone of scale regeneration; C, zone of osteoderm regeneration; D, site of the original tail with developed osteodermal cover. Scale bars: (**a**) = 3.5 mm, (**b**–**e**) = 2.5 mm, (**f**) = 500 µm.

**Figure 6 jdb-11-00022-f006:**
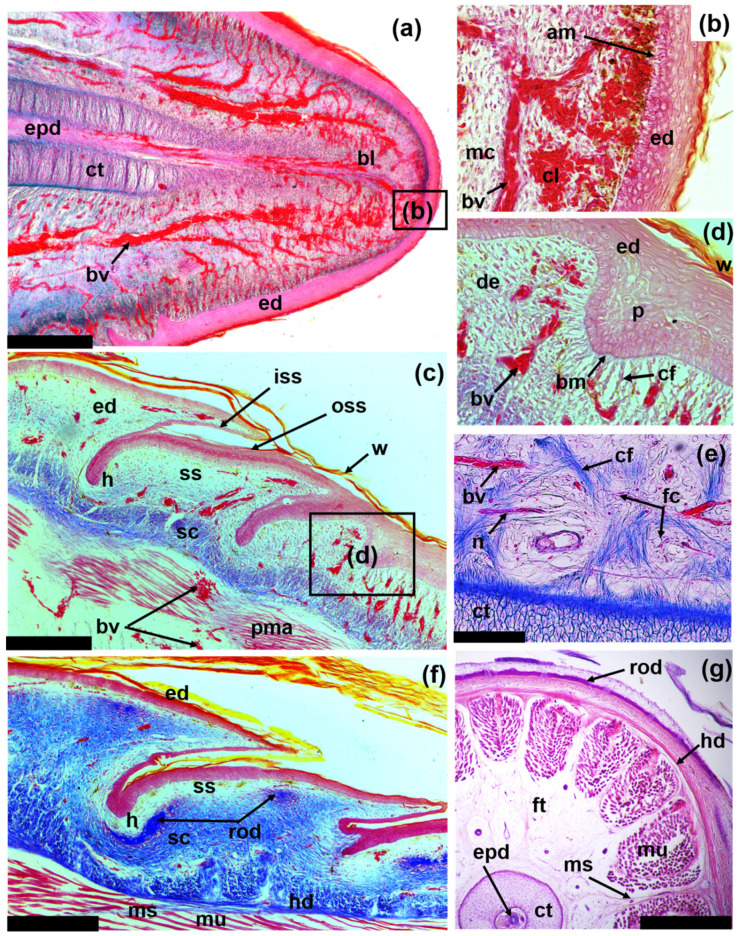
Frontal (**a**–**f**) and transverse (**g**) sections of regenerating tails of *E. taeniolata*: (**a**–**d**,**f**) regenerate 5.5 mm long (SPbU # 1775); (**e**) regenerate 37 mm long (SPbU # 1782); (**g**) regenerate 49 mm long (SPbU # 1780). Abbreviations: am, aggregates of melanocytes; bl, blastema; bm, basal membrane; bv, blood vessel; cf, collagen fibers; cl, clot; ct, cartilaginous tube; de, dermis; ed, epidermis; epd, ependyma; fc, fat cells; ft, fat tissue; h, hinge region between scales; hd, hypodermis; iss, inner scale surface; mc, mesenchymal cells; ms, myosepta; mu, muscles; n, nerve; oss, outer scale surface; p, peg; pma, promuscle aggregates; rod, regenerating osteoderm; sc, stratum compactum; ss, stratum superficiale; w, corneous layer of the wound epidermis. Sections are stained with Azan, azo-carmine B (**a**–**f**) and with Delafield’s hematoxylin, eosine (**g**). Scale bar: (**a**,**g**) = 500 µm, (**c**,**f**) = 250 µm, (**e**) = 100 µm. Internal structures of regenerating tail extend through final stage of development (**g**). Cartilaginous tube is composed of hypertrophied cells, which are covered inside and outside with perichondrium. Canal of cartilaginous tube contains an ependyma and several large blood vessels. Tube is surrounded by a thick layer of adipose tissue (**e**) and contains main blood vessels and spinal nerves. Muscles are well differentiated. Myomeres are separated by longitudinal and transverse myosepta composed of connective tissue fibers.

**Figure 7 jdb-11-00022-f007:**
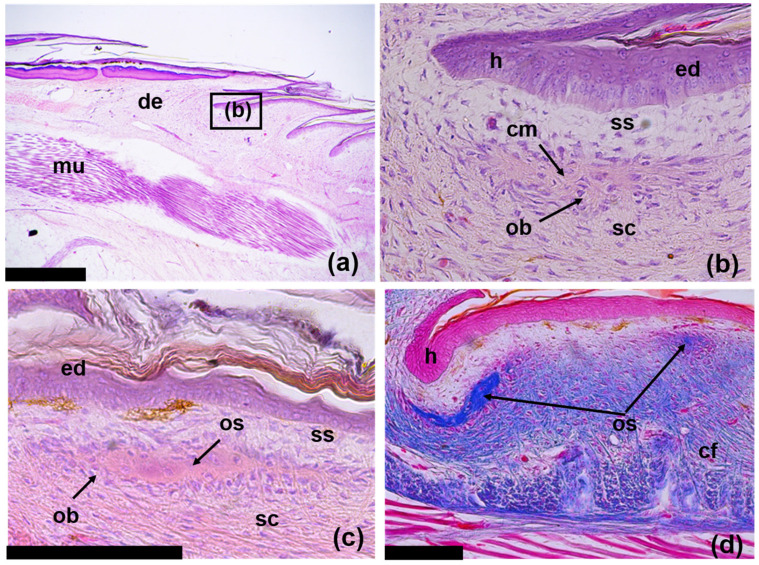
Longitudinal skin sections of regenerating tails of *E. taeniolata* in area of developing osteoderms: (**a**,**b**) regenerate 5 mm long (SPbU # 1773); (**c**) regenerate 7 mm long (SPbU # 1776); (**d**) regenerate 5.5 mm long (SPbU # 1775). Abbreviations: cf, collagen fibers; cm, collagen matrix; de, dermis; ed, epidermis; h, hinge region between scales; mu, muscles; ob, osteoblasts; os, osteoid; sc, stratum compactum; ss, stratum superficiale. Sections are stained with Delafield’s hematoxylin, eosine (**a**–**c**) and with Azan, azo-carmine B (**d**). Scale bars: (**a**) = 500 µm, (**c**,**d**) = 100 µm.

**Figure 8 jdb-11-00022-f008:**
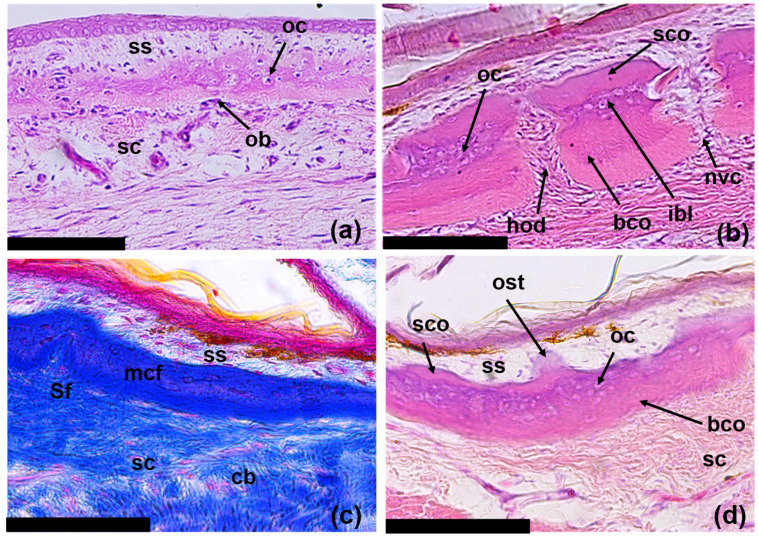
Transverse (**a**) and longitudinal (**b**–**d**) skin sections of regenerating tails of *E. taeniolata* in area of developing osteoderms: (**a**,**b**) regenerate 27 mm long (SPbU # 1779); (**c**,**d**) regenerate 37 mm long (SPbU # 1782). Abbreviations: bco, basal cortex; cb, collagen bundles; hod, hinge region between osteodrmites; ibl, intermediate bone layer; mcf, mineralized collagen fibers; nvc, neurovascular canal; ob, osteoblasts; oc, osteocytes; ost, osteodermine; sc, stratum compactum; sco, superficial cortex; Sf, Sharpey’s fibers; ss, stratum superficiale. Sections are stained with Delafield’s hematoxylin, eosine (**a**,**b**,**d**) and with Azan, azo-carmine B (**c**). Scale bars: 100 µm.

**Table 1 jdb-11-00022-t001:** List of histological preparations used in this study.

SPbU Numbers	Individual Numbers (Generation Numbers)	Regenerate Length (mm)	Regenerate Age (Month)	Character of Section	Number of Slides
1773	10 (1)	3	7.5	Longitudinal	3
1774	10 (2)	5	2	Longitudinal	8
1775	11 (1)	5.5	8	Longitudinal	6
1776	7 (1)	7	4	Longitudinal	7
1777	5 (1)	12.5	9.5	Longitudinal	8
1779	1 (2)	27	3.5	Transversal	3
				Longitudinal	10
1780	1 (1)	49	10	Transverse	1
1781	2 (1)	28	6	Longitudinal	13
1782	4 (1)	37	10	Longitudinal	5

**Table 2 jdb-11-00022-t002:** Parameters for scanning of specimens used in this study.

ZISP 18967 Numbers	Object to Source (mm)	Source Voltage (kV)	Source Current (uA)	Camera Exposure (ms)	Filter	Image Pixel Size (μm)
R-5328-2, male	63.071456	67	59	178	Al 0.5 mm	12.915298
R-5327-8, female	50.457308	58	68	127	Al 0.25 mm	10.332268
R-5334-5, female	36.040268	58	68	127	Al 0.25 mm	7.380055

## Data Availability

Not applicable.

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
