# Peer review of "Osteoderm Development during the Regeneration Process in Eurylepis taeniolata Blyth, 1854 (Scincidae, Sauria, Squamata)"

_jdb, 2023, doi:10.3390/jdb11020022_

Round 1

Reviewer 1 Report

Overall, this is an interesting and very useful contribution providing new details about osteoderm histology and regeneration in a species of skink.  Skinks are one of the most species-rich groups of lizards, and many/most skinks develop osteoderms, bony plates, within their skin.  The current contribution expands our existing knowledge about osteoderms and the process by which these elements can be replaced.

I only have one major comment: 

·      Field observations and maintaining individual lizards in captivity is reported in the Results, but the Institutional Review Board Protocols are not listed until the end of the paper – a reference to them in the Methods would be useful. That said, it is not clear if the lizards observed were used for subsequent data collection and/or if they were euthanized (and in which case, how were they euthanized?) or if they were returned to the wild. In addition, where did the field observations take place, what time of year, and were permits needed? How were the lizards maintained in captivity – details about enclosures, food, photoperiod, temperature, are all required.

Minor points for consideration

Introduction

·      Snakes (specifically sand boas) are now known to develop osteoderms – see Frydlova et al., 2023 (Scientific Reports 13: 6405)

·      In addition to TarentolaGeckolepis, and Gekko gecko, the gecko species Gekko reevesii also develops osteoderms – see Laver et al., 2020 (Journal of Morphology 281: 213-228)

Materials and Methods

·      Replace the term “preparations (glasses)” with “microscope slides” 

Results

·       Information related to the use of captive lizards required (noted above)

·       Figure 1 is a correlation analysis, but the data are not discussed – what do these findings reveal?

·       The genus/species Eurylepis taeniolata, as well as all other genera/species, need to be in italics

·       Perhaps note that anterior = cranial and posterior = caudal

·       Line 161: the entire osteoderm is made of compact bone, or just the basal plate?

·       The specimen in figure 4: how many days post-tail loss/regeneration is this individual?

·       Line 196: note that the zone of undifferentiated integument is the distal-most end of the tail. Further, the entire tail demonstrates a proximal to distal sequence of tissue differentiation/maturation. Related to this, without evidence of cell proliferation the term ‘blastema’ should be avoided (since cell proliferation is a key characteristic of the blasema. Focus on the undifferentiated state of the distal tip of the tail

·       Line 239: replace “immersion” with “invaginate”

·       Line 258: most authors do not include the hypodermis as part of the dermis – it is a separate layer of the skin

·       Line 276: are the chondrocytes vacuolated (is there evidence they have a vacuole) or just spherical?

Some genera/species need to be italicized; some minor editing but overall clear and easy to follow

Author Response

Manuscript ID jdb-2358009

Dear Editors, below are our replies to reviewers - Manuscript ID jdb-2358009
Thanks a lot for your cooperation.  
Reply to Reviewer_1: 
We are thankful for your effort to review our ms and for your valuable comments and corrections.
 We would like to answer to every point of your review:
I only have one major comment: 

1.  Field observations and maintaining individual lizards in captivity is reported in the Results, but the Institutional Review Board Protocols are not listed until the end of the paper – a reference to them in the Methods would be useful. That said, it is not clear if the lizards observed were used for subsequent data collection and/or if they were euthanized (and in which case, how were they euthanized?) or if they were returned to the wild. In addition, where did the field observations take place, what time of year, and were permits needed? How were the lizards maintained in captivity – details about enclosures, food, photoperiod, temperature, are all required.

The observations were carried out as part of cooperation and complex field work in the Turkmen SSR in 1987-1988, together with the Institute of Zoology of the Academy of Sciences of the Turkmen SSR (acknowledgements were made to the staff of this institute who our colleagues who already passed away). With this Institute in Turkmen SSR the Zoological Institute of Academy of Sciences of UdSSR (now Zoological Institute of Russian Academy of Sciences) and the Leningrad State University (now St. Petersburg) carried out a large-scale cooperation".  At present, this Turkmen institute has been disbanded. In those years, there were no ethical committees and, in the case of cooperation, permission to conduct research was not required. However, ethical standards for keeping animals were respected. In May 1987, 11 specimens of lizards were caught. For 12-14 months, the lizards were kept indoors in terrariums measuring 50 by 75 cm at a temperature of about 23-25º C (with gradient electric heating, which allowed the lizards to choose the optimal temperature regime) at a photoperiod corresponding to the geographical area of their habitat. The lizards were supplied with grasshoppers and crickets as food items, and water was available. After observations were made and material was obtained (tail regenerates at different stages of development), the lizards were released into their natural habitats. 

The graph shows the age of the regenerates, i.e. the time (number of months) of the regeneration process in a particular individual. It is calculated from the first autotomy to the second autotomy (1) or (in some individuals) from the second autotomy to the third (2). After autotomy, the tails were preserved.

Minor points for consideration

Introduction
1.•      Snakes (specifically sand boas) are now known to develop osteoderms – see Frydlova et al., 2023 (Scientific Reports 13: 6405)
- Yes, thanks, we agree, we got this paper available only after submission and were going to include this important reference. It will be included to the text of Introduction.  
•      In addition to Tarentola, Geckolepis, and Gekko gecko, the gecko species Gekko reevesii also develops osteoderms – see Laver et al., 2020 (Journal of Morphology 281: 213-228)
-    Thanks. We shall add this information.  

2.  Materials and Methods
•      Replace the term “preparations (glasses)” with “microscope slides”  - corrected

Results
•       Information related to the use of captive lizards required (noted above) – explanation is above 
•       Figure 1 is a correlation analysis, but the data are not discussed – what do these findings reveal?
It was partly discussed  the first paragraph of the "Results" section; now in accordance with reviewer 2 recommendations we moved it  to Materials section. This diagram of the relationship between size (mm) and age (month) of regenerating tails in Eurylepis taeniolata, showed that the rate of regeneration and size characteristics of tail regenerates depend on the individual characteristics of lizards. The first number indicates the number of the individual, in parentheses - the number of the regenerate, after the comma - the number of the corresponding histological preparation. 
       The genus/species Eurylepis taeniolata, as well as all other genera/species, need to be in italics – corrected
•       Perhaps note that anterior = cranial and posterior = caudal - corrected
•       Line 161: the entire osteoderm is made of compact bone, or just the basal plate?
Yes. The entire osteoderm is made of compact bone at the edges.
•       The specimen in figure 4: how many days post-tail loss/regeneration is this individual?
This specimen is 9.5 months old (was erroneously reported as length), 17.5 mm long. - corrected
•       Line 196: note that the zone of undifferentiated integument is the distal-most end of the tail. Further, the entire tail demonstrates a proximal to distal sequence of tissue differentiation/maturation. Related to this, without evidence of cell proliferation the term ‘blastema’ should be avoided (since cell proliferation is a key characteristic of the blasema. Focus on the undifferentiated state of the distal tip of the tail
Thank you! We made some corrections in the text where we meant not the blastema itself, but the area near the blastema
•       Line 239: replace “immersion” with “invaginate”  - corrected
•       Line 258: most authors do not include the hypodermis as part of the dermis – it is a separate layer of the skin
This is a debatable point. We prefer to keep the original text
•       Line 276: are the chondrocytes vacuolated (is there evidence they have a vacuole) or just spherical?
Replaced “vacuolated” with “hypertrophied” 
Comments on the Quality of English Language
Some genera/species need to be italicized; some minor editing but overall clear and easy to follow - corrected
Reply Date
10 May  2023
Date of this review
02 May 2023 18:56:54

Reviewer 2 Report

The authors provide a detailed description of the osteoderms in a skink species and discuss their regeneration process. Although the authors produced a lot of data, it is important to reorganise the text and figures such that it becomes more comprehensible to the reader. It is also possible to shorten it.

Here are some recommendations:

Introduction 

Some information from the discussion should be transferred to the introduction to better prepare the reader. For example, the fact that only skinks have compound ODs.

Methods

- the beginning of the results should be moved to the materials and methods such that the reader can immediately realise that living animals were sampled once or twice. 

- replace glass by slide

- what does 'series of sections' mean? number of sections on a slide? in the text '6 to 20 sections' and in the table '3 to 13'. Is this information important for the reader?

- explain why different stains were used. Do they put forward different relevant structures?

Results

- what do the numbers/codes in lines 120-121 correspond to?

- lines 144-149: refer to the relevant figure. Were there any measurements taken?

- line 157: relatively thin compared to what?

- line 166: provide information on Sharpey's fibres as they are important for the understanding of the results

- line 180-182: how is it possible to have less openings from the bottom?

- line 197: what does it 'and a small space near it'? Either it is the blastema area or not.

- line 242: what are the anchor filaments?

- line 256: in figures 4b-e, I don't see the continuous osteodermal cover.

- lines 263-266: where do we see the ossification centres?

- lines 283-294: refer appropriately to all panels of Fig. 6.

- line 302: where do we see the tuberculae?

- line 314-319: where do we see all this?

- line 322: were any statistical measurements performed to justify the term 'significantly'?

- lines 348-358: as suggested below, dedicate a paragraph/section on Figure 4 earlier in the Results or split it.

Discussion

- the beginning of the discussion belongs to the Introduction

- lines 379-383: to the Introduction

- line 385: what does it mean 'a high degree of probability' here?

- lines 408-420: nice description of the process. Use it as a guide to improve the results section.

Figures 

- It is necessary to add an image of the animal (intact tail) and an image of a regenerated tail. It will allow the reader to appreciate the difference in the shape of the original and regenerate scales which impact the shape and size of the ODs.

- Figure 3: (b) I don't see the difference between mcf and Sfb in the image.   (d) I don't see the difference between Sf and flb. Why different stains were used? What are expected to see?

- Figure 4: It is a mix of many different things and it is discussed at different parts of the manuscripts which is confusing. I suggest either to create a section and just describe this figure, or to split this figure and incorporate it in the other relavant figures.

- Figure 6: (a,b) I don't see the cf here.

- Figure 7: (c) improve the resolution and explain what we are supposed to see

- provide an overview image of the CTscan

It is important to shorten the manuscript by being more concise and avoid repetitions.

I strongly recommend to have a native English go through the text.

The binomial species names should always be in italics.

Author Response

Comments and Suggestions for Authors

Manuscript ID jdb-2358009

Dear Editors, below are our replies to reviewers - Manuscript ID jdb-2358009

The authors provide a detailed description of the osteoderms in a skink species and discuss their regeneration process. Although the authors produced a lot of data, it is important to reorganise the text and figures such that it becomes more comprehensible to the reader. It is also possible to shorten it.

Here are some recommendations:

Introduction 

Some information from the discussion should be transferred to the introduction to better prepare the reader. For example, the fact that only skinks have compound ODs. – We follow this recommendation.

Methods

- the beginning of the results should be moved to the materials and methods such that the reader can immediately realise that living animals were sampled once or twice. - We follow this recommendation.

- replace glass by slide – replaced

- what does 'series of sections' mean? number of sections on a slide? in the text '6 to 20 sections' and in the table '3 to 13'. Is this information important for the reader? –

64 preparations (slides) with 6 to 20 sections on a slide

- explain why different stains were used. Do they put forward different relevant structures? –

The azan/azacarmine staining method was used to visualize the structure of the collagen matrix.

Results

- what do the numbers/codes in lines 120-121 correspond to?

They correspond the abbreviations in the figure 1? Lcd  means the tail length consisting from two parts: original+regenerated

- lines 144-149: refer to the relevant figure. Were there any measurements taken?

lines 144-149: refer to the Figure 2b.  Due to the obvious difference in size. No measurements were taken.

- line 157: relatively thin compared to what?

We deleted “relatively”

 line 166: provide information on Sharpey's fibres as they are important for the understanding of the results

From Wikipedia: Sharpey's fibres are a matrix of connective tissue consisting of bundles of strong predominantly collagen fibres connecting periosteum to bone.

We provide the explanation in the text: «The basal cortex is connected to the underlying skin by bundles of collagen fibers (Sharpey’s fibers), deeply rooted in the bone and extending into the compact layer of the dermis.» -If it will be additional explanations needed?

- line 180-182: how is it possible to have less openings from the bottom?

This conclusion is made based on KT results 

- line 197: what does it 'and a small space near it'? Either it is the blastema area or not.

Corrections regarding the zone of the undifferentiated integument have been provided in the text. The blastema enters this zone as an integral part.

- line 242: what are the anchor filaments?

«The anchoring fibrils that link the lower part of the basement membrane to the papillary dermis are composed of type VII collagen» (Cell Biology (Third Edition), 2017).

- line 256: in figures 4b-e, I don't see the continuous osteodermal cover.

Thanks. We have made corrections - We have added a CT of a lizard with a fully developed tail regenerate (Figure 2).

lines 263-266: where do we see the ossification centres?

On the Figure 5d

- lines 283-294: refer appropriately to all panels of Fig. 6.

Two refers to the figures are added.

- line 302: where do we see the tuberculae?

The refer to the Figure 5c is added.

- line 314-319: where do we see all this?

They refer to the Figure 4a and Figure 4b.

- line 322: were any statistical measurements performed to justify the term 'significantly'?

corrected

- lines 348-358: as suggested below, dedicate a paragraph/section on Figure 4 earlier in the Results or split it.

We prefer to provide detailed explanations for the figure 4.

Discussion

- the beginning of the discussion belongs to the Introduction

- lines 379-383: to the Introduction

- line 385: what does it mean 'a high degree of probability' here? Corrected

- lines 408-420: nice description of the process. Use it as a guide to improve the results section.

On the basis of the extensive results obtained, this brief conclusion has been made

Figures 

- It is necessary to add an image of the animal (intact tail) and an image of a regenerated tail. It will allow the reader to appreciate the difference in the shape of the original and regenerate scales which impact the shape and size of the ODs

Many figures (CT, total preparation, histological sections) show areas with both regenerating scales and original ones. Thus, readers can evaluate the difference in the shape of the original and regenerated scales. We prefer to avoid to provide an additional figure..

- Figure 3: (b) I don't see the difference between mcf and Sfb in the image.   (d) I don't see the difference between Sf and flb. Why different stains were used? What are expected to see?

 (b) mcf, mineralised collagen fibres – redundant designation, replaced by flb, fibrolamellar (parallel-fibred) bone.

 (d) In this case, the differences are significant: Sf, Sharpey's fibers are fibers located in the dermis, and Sfb, Sharpey-fibre bone are fibers included in the bone. The staining method allows us to clearly visualize the structure of the collagen matrix (this is important here), but obscures the boundary between the collagen of the dermis and the bone.

- Figure 4: It is a mix of many different things and it is discussed at different parts of the manuscripts which is confusing. I suggest either to create a section and just describe this figure, or to split this figure and incorporate it in the other relavant figures.

Figure 4 focuses on images of total preparations, which demonstrate the difference in the structure of the original osteodermal cover and the regenerating one at different stages of development. This complex  figure  allows us to visually compare these structures. Therefore, we prefer to keep it.

- Figure 6: (a,b) I don't see the cf here.

This is the initial stage of development of the regenerate 5 mm long. Collagen fibers here are very thin, visible only at high magnification. Nevertheless, in the region of the osteoderm rudiment (b), a condensation of the collagen matrix is visible.

- Figure 7: (c) improve the resolution and explain what we are supposed to see

This section is stained with Azan, azocarmine. It allows readers to see the structures of collagen fibers in the dermis and bone tissue.  They are visible under this resolution. 

- provide an overview image of the CTscan  done

Comments on the Quality of English Language

It is important to shorten the manuscript by being more concise and avoid repetitions. Done.  We deleted some sentences and paragraphs.

I strongly recommend to have a native English go through the text – Done. native English speaking colleagues  Kraig Adler, Cornell University, USA and Jesse L. Grismer, La Sierra University send  his corrections. Corrections are incorporated.

The binomial species names should always be in italics. -corrected

Reply Date

11 May 2023

Date of this review

20 Apr 2023 16:24:24

Round 2

Reviewer 2 Report

The authors provide an improved version of their manuscript which clearly describes their findings.

I only have a few minor comments/corrections:

- line 33: remove only

- line 34: in snakes of the Erycinae

- lines 42-47: repetition

- line 51: remove quotes

- lines 106-119 and Figure 1: this part belongs to the Results section

- line 140: please explanation of Lcd in the main text

- line 202: remove 'of areas'

- lines 202-205: refer to figure

- line 214: refer to figure 5a, instead of figure 5 in general

- line 250: add figure reference

- line 264: is reference to Figure 7a relevant here?

- line 314: repetition

- line 380: remove 'referred' repetition

- lines 407-420: mention these stages in the results section

Author Response

Dear editors,
Dear reviewer,
Thanks a lot for your cooperation and your valuable contribution to the editorial work 
with our manuscript.
Below replies to all the comments/corrections:
- line 33: remove only - removed
- line 34: in snakes of the Erycinae - corrected
- lines 42-47: repetition -corrected
- line 51: remove quotes -removed
- lines 106-119 and Figure 1: this part belongs to the Results section -transferred
- line 140: please explanation of Lcd in the main text provide this explanation
- line 202: remove 'of areas' - removed
- lines 202-205: refer to figure provide the reference
- line 214: refer to figure 5a, instead of figure 5 in general -corrected
- line 250: add figure reference - added
- line 264: is reference to Figure 7a relevant here? corrected
- line 314: repetition -corrected
- line 380: remove 'referred' repetition -removed
- lines 407-420: mention these stages in the results section transferred
Date of reply   May 17th
